# Morphological and Molecular Characterization of *Lema bilineata* (Germar), a New Alien Invasive Leaf Beetle for Europe, with Notes on the Related Species *Lema daturaphila* Kogan & Goeden

**DOI:** 10.3390/insects11050295

**Published:** 2020-05-11

**Authors:** Maurilia M. Monti, Michelina Ruocco, Elizabeth Grobbelaar, Paolo A. Pedata

**Affiliations:** 1National Research Council (CNR), Institute for Sustainable Plant Protection, 80055 Portici, Italy; michelina.ruocco@ipsp.cnr.it; 2Agricultural Research Council (ARC), Plant Protection Research Institute (PPRI), Private Bag X134, Queenswood, Pretoria 0121, Gauteng, South Africa; GrobbelaarB@arc.agric.za

**Keywords:** COI, color pattern, Criocerinae, genitalia, Solanaceae

## Abstract

*Lema bilineata* (Germar) is an alien invasive leaf beetle (Coleoptera: Chrysomelidae) first recorded in Europe in the summer of 2017 in the province of Naples (Campania, Italy). It occurs on both cultivated plants (*Nicotiana tabacum*) and weeds (*Salpichroa origanifolia* and *Datura* spp.). Information on morphological characters, color variation and molecular data are deficient for *L. bilineata*, as is the case for most *Lema* species. These data could be useful to discriminate between this species and the closely related *Lema daturaphila* Kogan & Goeden, which has the same potential to become an alien invasive species. In this paper, color variation in adults and the morphology of the aedeagi and spermathecae of the two species are documented and compared, including micrographic images. Additional data on the current distribution of *L. bilineata* in Campania is also provided. The cytochrome c oxidase I (COI) barcoding region of both Italian and South African specimens of *L. bilineata*, as well as South African specimens of *L. daturaphila*, was sequenced. A preliminary phylogenetic tree is provided, based on the sequences available for *Lema* species.

## 1. Introduction

In July–August 2017 the leaf beetle called the Tobacco Slug, *Lema bilineata* (Germar, 1823) (Coleoptera: Chrysomelidae: Criocerinae), was found on *Physalis peruviana* L. and subsequently on *Salpichroa origanifolia* (Lam.) Baill., in the province of Naples (Campania, Italy). This finding was the first record for this species in Europe [1]. *Lema bilineata* is a species native to South America [2,3], which became invasive in South Africa, having probably been imported from Argentina in horse fodder during the Anglo Boer War [4] and successively spread to Zimbabwe [5] and Australia [6]. *Lema bilineata* is a moderately oligophagous species, feeding on host plants belonging exclusively to the family Solanaceae, with some other dubious exceptions reported. In its native range, it appears to be associated mainly with *S. origanifolia* [7]. The only crop attacked by *L. bilineata* is *Nicotiana tabacum* L., to which the beetle has been reported to cause economic losses both in Argentina [8] and South Africa [9]. In contrast, in Australia where the growing of tobacco is illegal, it is not considered an economic pest, but rather valued as a potential biological control agent of weeds [6].

Despite its economic importance, most of the available data on *L. bilineata* are related to its feeding preferences [10,11,12,13]. Since little information on color variation is available for this species [14,15], detailed morphological, anatomical and molecular data are needed. A deeper characterization of this pest is important to effectively distinguish it from related species, such as *Lema daturaphila* Kogan & Goeden, 1970, which has a similar host range and invasion history ([16,17] both as *L. trilineata*; [9] as *L. trilinea*). However, the taxonomy of *Lema* species, and Criocerinae in general, is problematic, due to a relatively uniform body shape and lack of reliable diagnostic characters. Many descriptions are based mainly on color, even though this character may display considerable intraspecific variation. Such variation may also be difficult to assess, as collection of a large number of specimens from the same site is not always possible [18]. In addition to color, anatomical characteristics, such as the genitalia, have proven to be useful in the taxonomy of this group, but they have only been described for a few species [19].

In this study we collected color, anatomical and molecular data on *L. bilineata* and *L. daturaphila* that will be useful to distinguish these species. Such data can be used to monitor these beetles more efficiently at customs check points, to prevent potential new introductions, and to promptly implement eradication measures for new alien invasive species subsequent to field inspections. Furthermore, the distribution of *L. bilineata* in the provinces of Campania was assessed through regular samplings on tobacco and other host plants, and acceptance of alternative hosts was evaluated.

## 2. Materials and Methods

### 2.1. Field Monitoring in Campania

During 2018–2019 the main tobacco growing districts in Campania were regularly monitored through visual inspections, to detect the presence of both adults and pre-imaginal stages of *L. bilineata*. Pest management of monitored fields was implemented within the current strategies to control aphids and *Epitrix hirtipennis*, using pyrethroids and neonicotinoids. Host weeds, such as *Datura* spp. and *S. origanifolia*, accidentally occurring in the nearby tobacco plantings, were also inspected.

### 2.2. Color and Anatomical Observations

The color pattern of *L. bilineata* was observed on hundreds of specimens, both collected in several open fields in Campania and reared in the laboratory, as well as on 321 museum specimens from the South African National Collection of Insects (SANC). Color patterns for *L. daturaphila* were observed on 225 museum specimens from the SANC.

Live individuals of *L. bilineata* collected in Campania were killed with ethyl acetate vapor and immediately prepared for observation. Museum specimens of *L. daturaphila* and of the extreme color variations of *L. bilineata* were first softened and cleaned before remounting to facilitate photography. Photographs were taken with a Zeiss Axiophot 2 microscope equipped with an Axiocam HRC digital camera (Carl Zeiss, Oberkochen, Germany), and fully-focused images were obtained by combining multiple planes of focus by CombineZP^®^ software (2 May–30 June 2018) (CNR) and a motorized Zeiss Discovery V.20 SteREO microscope equipped with an Axiocam 506 colour camera (Carl Zeiss, Oberkochen, Germany), and stacked using Helicon Focus 7.5.8 software (1 October–29 November 2019) (ARC).

For observation of the aedeagus and spermatheca we used several freshly killed male and female specimens of *L. bilineata*, as well as museum specimens of *L. bilineata* and *L. daturaphila* from the SANC.

After softening dry specimens in boiling water, the abdomen was removed and soaked in a 10% potassium hydroxide solution at 50 °C overnight. The aedeagus and spermatheca were then extracted and rinsed in 70% ethanol with a drop of glacial acetic acid to neutralize the potassium hydroxide, followed by a rinse in pure 70% ethanol, before being mounted in glycerin for observation. Photographs were taken using a motorized Zeiss Discovery V.20 SteREO microscope, Axiocam 506 colour camera and Z-stacked using Zen 2.3 Pro Software (2 May–30 June 2018) (ARC).

Terminology for male genitalia is based largely on White [20]. Descriptions of the internal-sac sclerites follow the terminology established by Matsumura and Yoshizawa [21] who named sclerites based on their position in repose. Description of the spermatheca follows Matsumura et al. [22].

### 2.3. Molecular Data

For molecular analysis we used specimens of *L. bilineata* collected at different sites in Campania (Italy). In addition, we examined South African specimens both of *L. bilineata* and *L. daturaphila* from the SANC (Table 1). DNA was extracted from a single tarsus using the modified Chelex method [23]. The cytochrome c oxidase I (COI) barcoding region was amplified from fresh specimens with primer pair LCO-1490/HCO-2198 [24]. PCR condition was as follows: 94 °C for 50 s followed by 40 cycles of 94 °C for 30 s, 48 °C for 40 s, 72 °C for 60 s, with a final elongation step at 72 °C for 10 min. As the DNA of dry museum specimens was fragmented (making it impossible to amplify the barcoding region of COI gene with the above primers), internal primers MH-MR1 and MF1 [25] coupled with LCO and HCO, respectively, were used. A step-up amplification was set up, with 94 °C for 50 s followed by 5 cycles of 94 °C for 30 s, 46 °C for 40 s, 72 °C for 60 s, and 35 cycles of 94 °C for 30 s, 48 °C for 40 s, 72 °C for 60 s, and a final elongation step at 72 °C for 10 min. PCR products were purified and directly sequenced on both strands using the marker-specific primers. 

Chromatograms were assembled using BioEdit 7.0 [26] and edited manually. Spurious ampli fications of COI sequences were checked using Standard Nucleotide BLAST. Alignment of sequences was performed with ClustalW [27], and COI sequences were verified for protein-coding frameshifts, nonsense codons, indels, and stop codons to avoid the inclusion of possible pseudogenes [28,29]. All novel nucleotide sequences of COI genes were deposited in the GenBank database.

The data set for the evolutionary analyses includes 20 new sequences (17 of *L. bilineata* and 3 of *L. daturaphila*) and sequences of other 14 identified *Lema* species and 3 unclassified *Lema* species, downloaded from GenBank (last access on 16 December 2019) and the Barcode of Life Data (BOLD) System [30]. Intra- and interspecific pairwise distances were calculated using MEGA X [31] implementing the K2P model (Kimura-two-parameter) [32]. The phylogenetic tree was reconstructed using maximum likelihood (ML) inference [33], implementing the GTR + G + I evolutionary model [34] as selected by jModeltest [35]. ML branch support was based on 10,000 rapid bootstrap pseudoreplicates to evaluate the branching confidence and clades were considered supported when bootstrap was >70%. *Lilioceris lilii* Scopoli was chosen as the outgroup to root the COI tree. Phylogenetic analyses were performed with MEGA X software.

### 2.4. Host Plant Range

To evaluate the host plant range of *L. bilineata* the following species (many of which are mentioned in the literature as host plants) were tested: *Nicotiana glauca* Graham and *N. glutinosa* L., *Brugmansia arborea* (L.) Sweet, *Datura stramonium* L., *Petunia hybrida* Vilm., *Solanum tuberosum* L. and *S. lycopersicum* L., *Capsicum annuum* L. (glabrous and pubescent varieties) (all Solanaceae), and among non-solanaceous species, *Helianthus annuus* L. (Asteraceae), *Beta vulgaris* L. (Amaranthaceae)*,* and *Ribes grossularia* L. (Grossulariaceae). Each plant species was assessed in no choice tests with adults and larvae. Five replicas were carried out for each sampled species, and each replication consisted of 5 newly emerged adults or 5 egg masses (of about 20 eggs). Feeding activity for both life stages of the beetle was assessed for 7 consecutive days.

## 3. Results

### 3.1. Field Monitoring in Campania

In 2018 *L. bilineata* expanded its invasion range and was found on *N. tabacum* in most of the tobacco growing districts of Campania (Figure 1), with the exception of the province of Avellino (where the tobacco growing is very limited) and most of the sampled sites in the province of Benevento, where the main tobacco production (both quantitative and qualitative) of Campania is located. The most severe damage was caused in the provinces of Naples and Caserta, with peaks of 25% of attacked leaves during July–August. In all the other inspected fields the pest was found sporadically. During 2019, *L. bilineata*, even though still common on *S. origanifolia* in the provinces of Naples and Caserta, was only present at negligible levels on tobacco, while it was absent in the most important districts in Benevento province.

### 3.2. Color and Anatomical Observations

***Lema bilineata*** (Figure 2; Figure 3a,b,e; Appendix A)

***Size***: average total length 7.6 mm (s.d. 0.53), average width at humeral callus 3.2 mm (s.d. 0.13) (n = 10).

***Dorsum*** (Figure 2): head entirely black or with a yellowish- to reddish-brown area between the eyes and basally. Antenna entirely black or antennomeres 1 to 5 sometimes, and 10 and 11 rarely, partially yellowish- to reddish-brown. Pronotum yellowish- to reddish-brown, sometimes with a small amount of black medially at the base which may extend around to the ventral side; generally, with two irregular black spots, one on each side towards the anterior third; often enlarged to form two large rectangular black areas that cover most of the pronotum; sometimes entirely black. Scutellum always black. Elytron usually yellowish-brown, sometimes a paler creamy-yellow, with a broad black lateral longitudinal stripe extending from the humeral callus to just before the apex; the sutural margin broadly bordered with black from base to apex, extending a short distance onto the outer elytral margin; the elytra together appear to have two broad black lateral longitudinal lines encircled by a pale creamy-yellow to yellowish-brown area, and a broad black median line.

***Venter*** (Figure 2): head from yellowish- to reddish-brown, to entirely black; if largely black sometimes yellowish- to reddish-brown medio-basally. Prosternum yellowish- or reddish-brown to entirely black, sometimes with varying amounts of black laterally and towards the base, rarely joining the irregular black dorsal spots; meso- and metathoracic sternites ranging from a yellowish- to reddish-brown background with a dark area towards the base of the metasternum (rarely absent), to entirely black; dorso-lateral areas on sclerites of the meso- and metathorax mostly black. Legs from yellowish-brown with various black markings to entirely black; coxae mostly black with varying amounts of yellowish- to reddish-brown, to entirely black; trochanters from reddish-brown to black; femora black with basal quarter to half yellowish- to reddish-brown and sometimes with a small circular yellowish- to reddish-brown spot on the outside near the apex, to entirely black; femora rarely largely yellowish- to reddish-brown, with a small amount of black in the apical third, sometimes in the form of spots anteriorly and posteriorly, which may be joined ventrally; tibiae yellowish- to reddish-brown, sometimes with black on the outside, but always slightly to distinctly black apically, to entirely black; tarsi generally black, but yellowish- to reddish-brown basally in a few specimens. Abdomen yellowish- or reddish-brown, usually with variable black markings, to completely black; first ventrite entirely yellowish- to reddish-brown, sometimes black apically and dorso-laterally, often with a medial almost diamond shaped black area towards the apex, to entirely black; ventrites 2–4 range from entirely yellowish- to reddish-brown with varying amounts of black medially at the base and laterally; lateral areas often extend apicad from the base, but are sometimes reduced and only form latero-medial spots; the medial and lateral black areas sometimes merge to form transverse stripes that extend across most of the ventrite width; fifth ventrite black medially towards the apex; abdomen rarely black with yellowish- to reddish-brown on the baso-lateral areas of the first ventrite; or entirely black, then often with two small yellowish-brown areas on the outer extremes of the fifth ventrite. 

***Aedeagus*** (Figure 3a,b): (n = 15) in lateral view, greatest width towards the apex; apex sharply pointed and directed forward, with the dorsal apical margin straight; in dorsal view, widening slightly towards the apical section; orifice overlaid by three lobes, a large median lobe and two smaller lateral lobes; the internal profile, anteriad of the orifice, rounded; internal processes symmetrical in dorsal view; dorsal sclerite of the internal sac about 0.75× the length of the ventral sclerite. Internal sac apparently without flagellum and pocket. 

***Spermatheca*** (Figure 3e): (n = 10) spermathecal capsule relatively strongly sclerotized and brown, elongate with distinct coils (except one specimen) proximally and deeply hook-shaped distally.

***Color pattern***: relatively constant on the elytra, but the pronotum and the ventral region show considerable variation. The pronotum varies from having two irregular black spots, one on either side towards the anterior third, on a yellowish- to reddish-brown background, to entirely black; most commonly the two spots are replaced by two large rectangular black areas that cover most of the pronotum (these are joined medially in one specimen and basally in another); in one specimen the black pronotum has an irregularly yellowish-brown anterior margin with a medial spot of the same color in the apical third and a small transverse medial area of the same color in the posterior third; a few specimens have one or two variously placed small yellow areas, either medially or one on either side, on an otherwise black pronotum. On the elytra two broad black lateral longitudinal lines encircled by a pale creamy-yellow to yellowish-brown area are constant, except in a few individuals, where the pattern may assume a darker or paler color. In darker specimens, the elytra are almost entirely black and we found: one specimen (SANC) with only the medial part of the lateral elytral margin yellowish-brown; four specimens (three SANC, one Italy) with yellowish-brown areas only faintly visible towards the apex, the middle, and the extreme lateral margins of the elytra; one specimen (SANC) with the broad black sutural margin joining the broad black lateral margin medially. Among paler specimens we found eight individuals (SANC) with the black stripes barely visible, the elytron almost entirely yellowish-brown with the sutural margin narrowly bordered with indistinct to distinct black from about one-fifth from the base to the apex, terminating in an indistinct to distinct black spot in the apical angle; one of these specimens also has an indistinct black spot on the humeral callus. The ventral side ranges from almost completely yellowish- to reddish-brown with a dark area on the posterior part of the metasternum and small amounts of black on the dorso-lateral thoracic sclerites and barely visible black dorso-lateral spots on ventrites, to black with a small yellowish- to reddish-brown area medially on the mesosternum, to entirely black. In a few Italian specimens the ventral side is completely reddish-brown.

Even if *L. bilineata* presents remarkable color variation, the pronotum, the ventral side and the legs usually share a common pattern. Indeed, a pronotum with a reddish-brown background is always associated with a ventral side that is at least partially reddish-brown (except for rare extreme cases), while generally in specimens with completely black pronotum, the ventral side (including legs) is also totally black, except for few individuals with very little to no yellowish- to reddish-brown on the medial anterior margin of the prosternum, between the meso-coxae and on the outer extremes of the fifth ventrite. Rarest forms, with almost entirely black elytra, are not always associated with the darkest coloration of the pronotum and ventral side, whereas those with elytra with almost no markings are not always associated with the pronotum and ventral side with the smallest black markings. Moreover, freshly emerged adults might present a clearer coloration without the black spot on the metasternum and excessively yellow legs, including the coxae, before assuming the definitive color pattern. Finally, there are a few specimens in the SANC collection, classified as *L. bilineata*, showing extreme color variation not found among the Italian material (Figure 2 in frame): a pale yellowish-brown form, showing only an incomplete dark brown bordering of the sutural margin and the usual dark spots on pronotum, and a completely black form (both on dorsum and venter). One of the pale specimens was collected together with three specimens showing typical coloration on 14.v.1920 in Durban, KwaZulu-Natal by C.P. van der Merwe (SANC accession number AcC2822); the completely black specimen was collected with three other dark specimens on ii.1921 on tobacco in Umbilo, KwaZulu-Natal (SANC accession number AcSN3695).

***Lema daturaphila*** (Figure 4; Figure 3c,d,f; Appendix A)

***Size***: average total length 7.7 mm (s.d. 0.53), average width at humeral callus 3.2 mm (s.d. 0.19) (n = 10).

***Dorsum*** (Figure 4): head entirely yellowish-brown to entirely black, sometimes yellowish-brown with a variable number and size of black areas near the base, or black basally; inner and dorsal ocular margin black; sometimes black towards the anterior of the head. Antenna entirely black, sometimes antennomeres 1–2 partially, or entirely, yellowish-brown. Pronotum yellowish-brown, generally with two small black spots, one on each side towards the anterior third; sometimes with an additional elongate medial spot; spots rarely greatly enlarged forming rectangular patches which cover most of the anterior two-thirds of the pronotum and are sometimes joined basally by a small medial spot. Scutellum always black. Elytron yellowish-brown with a narrow black longitudinal stripe extending from the humeral callus to just before the apex; the sutural margin narrowly bordered with black from base to apex; the elytra together appear to have three narrow black longitudinal lines.

***Venter*** (Figure 4): head yellowish-brown to black with traces of yellowish-brown towards the apex. Prosternum yellowish-brown; meso- and metathoracic sternites yellowish-brown, sometimes black towards the posterior of the metathorax; rarely black with only the prosternum and medial section of the mesosternum yellowish-brown; black areas variable on the dorso-lateral areas of sclerites of the meso- and metathorax. Legs yellowish-brown, usually with black areas on the coxae; trochanters yellowish-brown, sometimes suffused to a greater or lesser extent with black; femora generally black on apices and sometimes dorsally; tibiae yellowish-brown with apices, or entire outer surface, black, or entirely black; tarsi black; legs may be entirely black with only the basal quarter to third of the femora yellowish-brown. Abdominal ventrites entirely yellowish-brown; sometimes with an apico-medial black area, varying in size, on the first ventrite, sometimes together with a similar spot on the second, third and fourth ventrites; rarely also triangular black marks on the apico-lateral margin of the two basal ventrites; abdomen rarely black, then with only the apico-lateral margins of the first four ventrites and the apical ventrite yellowish-brown.

***Aedeagus*** (Figure 3c,d): (n = 10) in lateral view, greatest width towards the apex; apex sharply pointed and directed forward, with the dorsal apical margin slightly concave; in dorsal view not widening significantly towards the apical section; orifice overlaid by three lobes, a large median lobe and two smaller lateral lobes; the internal profile, anteriad of the orifice, sub-triangular in shape; internal sclerites symmetrical in dorsal view; the dorsal sclerite of the internal sac about 0.5× the ventral one. Internal sac apparently without flagellum and pocket.

***Spermatheca*** (Figure 3f): (n = 10) spermathecal capsule relatively strongly sclerotized and brown, elongated with slight to distinct coils proximally and shallowly hook-shaped distally.

***Color pattern***: relatively constant on the elytra, while the head, pronotum, legs and the ventral region show much variation. The pronotum displays anything from no black spots, to having varied black spots/areas on either side in the anterior third; these may rarely form two large black rectangular patches; in the most extreme case, one specimen has a single rectangular patch covering most of the anterior two-thirds of the pronotum; most commonly there are two black spots, one on either side in the anterior third. The elytra are always yellowish-brown, together displaying three narrow black longitudinal lines; only two specimens have no lateral lines and the sutural margins are bordered with black from about one-fifth from the base to the apex, terminating in a black spot in the apical angle. The ventral side is mostly yellowish-brown, sometimes with a varying amount of black on the posterior area of the metathorax, the dorso-lateral sclerites of the meso- and metathorax, and the ventrites. In one specimen, the head, antennae, legs and most of the meso- and metathoracic and segments and ventrites are black, only the prothorax is yellowish-brown with a large black area covering most of the anterior two-thirds of the pronotum; the trochanters and basal fifth of the femora are yellowish-brown; and the medial section of the third and fourth ventrite and the apical ventrite are yellowish-brown. Excessive melanic coloration is the exception rather than the rule.

### 3.3. Molecular Analyses

COI of 7 individuals of *L. bilineata* from Italy and 10 from South Africa and 3 individuals of *L. daturaphila* from South Africa were sequenced. After trimming for incomplete ends, we obtained 562 unambiguously aligned bp sequences, without frame shifts or nonsense codons. These sequences were compared with those of *Lema* species available in public databases. Due to the difference in length when compared to published sequences, the complete data set was trimmed to 382 bp. 

Phylogenetic analyses recovered *L. trivittata* Say and *L. daturaphila* (including specimens reported with the invalid name *L*. *trilinea*) as the closest related species of *L. bilineata*. The COI tree showed that *L. bilineata* and *L. daturaphila* belong to two highly supported monophyletic clades (bootstrap values 99%) (Figure 5).

The mean intraspecific K2P nucleotide distance among all *L. bilineata* sequences was 0.73%, with distances ranging from 0% to 1.62% among South African samples and from 0% to 0.80% among Italian samples. Among *L. bilineata* sequences, Italian and South African specimens share one haplotype. 

The mean intraspecific K2P nucleotide distance within *L. daturaphila* was 0.94%. This species is split in two different highly supported clusters (both with 81% bootstrap value), one with samples from North-Central America, with distances ranging from 0% to 0.80%, and one with samples from South Africa and Australia, with distances ranging from 0% to 0.27%. In the first *L. daturaphila* clade, two not-identified sequences were also recovered, mined from the BOLD database. This result confirms these specimens belong to the *L. daturaphila* cluster (BIN BOLD: AAG4425) as reported in the BOLD database. 

The dataset of *L. bilineata* and *L. daturaphila* showed an average base composition of A 30.7%, T 37.9%, G 14.8%, and C 16.6%, and A 30.1%, T 38.4%, G 14.6%, and C 17.0%, respectively, both close to the average composition reported for Chrysomelidae [36]. 

Nucleotide differences among samples were predominantly in the third-codon position and synonyms. The interspecific distance between *L. bilineata* and *L. daturaphila* was 12.08%, and the intraspecific variation did not overlap with the variation between species; these data are consistent with results of intra- and interspecific variabilities reported for Criocerinae [36].

### 3.4. Host Plant Range

Both adults and larvae readily accepted *N. glauca* and *D. stramonium* as alternative hosts to *N. tabacum, P. peruviana* and *S. origanifolia,* and were able to complete their development on these host plants. *Nicotiana glutinosa* and *B. arborea* were accepted by adults, but not by larvae, as were *P. hybrida* and glabrous varieties of *C. annum*, on which adults caused minor lesions. All other host plant species tested were not accepted by either the adults or larvae.

## 4. Discussion

*Lema bilineata* was widespread in many tobacco producing districts of Campania during the monitoring period, causing damage mostly in the first year of its appearance, while during 2019 it did not cause any significant economic loss to tobacco crops. Invasion of new territories by alien invasive species is an increasing global phenomenon, which poses serious risks to both natural and agricultural ecosystems, with a resulting loss of biodiversity and an increase in the use of pesticides. To reduce the threat of new introductions, a crucial step is the development of efficient tools for accurate identification and early detection of invasive species.

The alien invasive species *L. bilineata*, recently introduced in Italy, is characterized by a high phenotypic plasticity, particularly in its color pattern, which can overlap with that of two closely related species, *L. daturaphila* and *L. trivittata*. These three species belong to a group of species feeding on Solanaceae [18], but each has a different host range and damage potential.

The most frequently observed color pattern for *L. bilineata* shows the elytra with broad black lateral longitudinal lines encircled by a pale creamy-yellow to yellowish-brown area, and a third broad black longitudinal line formed by the sutural margins that extends from the base to the apex and a short distance onto the outer elytral margin; the pronotum with two black spots on a yellowish- to reddish-brown background, and the ventral side yellowish- to reddish-brown with black areas on the pro-, meso- and metasternites, as well as the dorsal areas of the lateral thoracic sclerites. However, we have also found a less common form, with an entirely black pronotum and ventral side, and a rare form with almost entirely black elytra, which is not always associated with black pronotum and ventral side. An even greater variation has been observed in specimens of the SANC collection, with patterns ranging from largely yellowish-brown to completely dark. Our observations on specimens of *L. bilineata* sampled in southern Italy and in specimens from the SANC collection confirm the great variability in coloration reported by other authors [14,15], even if the extreme dark form with black elytra has not yet been reported. Heinze and Pinsdorf [14] reported a pale form with completely creamy-yellowish to yellowish-brown elytra, described as an aberrant form of *L. bilineata* with the name ab. *flavipennis* Heinze, opposed to the dark form, with a completely black pronotum, described as an aberrant form with the name ab. *nigricollis* Heinze. The pale form of the SANC specimen could be consistent with the form *flavipennis*.

*Lema daturaphila* is a species that originates from North and Central America, reported for the first time out of its native range in South Africa as an economic pest of Cape Gooseberry (*P. peruviana*) [16] and subsequently on tobacco [9]. In the late 1970s it was recorded in Australia on potato and other Solanaceae [17]. The taxonomic history of *L. daturaphila* is problematic [20,37,38], as reflected in the different names used for this species in the literature, such as *L. trilineata* [16,17], *L. trilinea* [9] and even *L. trivittata* [6]. *Lema trivittata* is a North American species, morphologically very similar to *L. daturaphila* [37] and associated, among other hosts, with *S. tuberosum* [39], which was recently found on some Japanese islands [40,41] and in Taiwan [42]. *Lema daturaphila* and *L. trivittata* both have largely yellow elytra with three narrow black stripes, and a yellowish-brown pronotum with two black spots. However, melanic forms have been reported from the eastern USA for *L. daturaphila*, which have an elytral color pattern resembling that of *L. bilineata* [20]. On the other hand, paler forms of both species, observed in our samples, exhibit considerable overlap in their color patterns.

Therefore, even if the most common color pattern of *L. bilineata* is clearly distinct from that of *L. daturaphila*, the use of this character on its own may result in an incorrect identification when separating the specimens exhibiting the extreme color variations. From this perspective, the identification of additional characters may be valuable when dealing with ambiguous cases.

Male genitalia have proved to be of taxonomic value in many insect groups, including the Chrysomelidae, because of their rapid divergence due to sexual selection [43]. This feature makes them a useful character to distinguish between even closely related species, but their efficacy must be established for any specific taxon. For example, White [20] considered the aedeagus of limited value for distinguishing *Lema* species, but useful within the genus *Oulema*. Even though this character is quite similar in *L. bilineata* and *L. daturaphila*, some differences can be highlighted: in lateral view, in *L. bilineata* the dorsal apical margin is straight while in *L. daturaphila* it is slightly concave; in dorsal view, the internal profile anteriad of the orifice is rounded in *L. bilineata*, but sub-triangular in *L. daturaphila*; the ratio of dorsal:ventral sclerites of the internal sac is higher in *L. bilineata* (0.75) than in *L. daturaphila* (0.5). Both species have an internal sac apparently devoid of a flagellum and pocket, which appears to be the plesiomorphic state for the genus [22]. A definitive proof of the absence of flagellum could be gained through observation of everted endophallus [44].

Internal female reproductive organs in Chrysomelidae, in particular the spermatheca, allow discrimination at generic and specific levels, although less frequently than male genitalia [22,45,46]. *Lema daturaphila* and *L. bilineata* have similar spermathecal capsules which are elongated and coiled proximally with a hook-shaped distal part; coils in the proximal section show overlapping in the degree of convolutions in the two species, therefore this character is not suitable to discriminate them. This kind of spermatheca is a synapomorphy of the *Lema* + *Lilioceris* group, even though this character successively evolved different shapes, including reversal, among *Lema* lines [22].

As far as we know, this is the first study regarding the molecular identification of *L. bilineata*. At species level, phylogenetic analysis confirms that *L. bilineata* and *L. daturaphila*, even if they might show overlapping morphological characters, are reliably identifiable with COI markers. These two species, indeed, belong to two different and highly supported monophyletic clades (bootstrap value: 99%) and can be clearly separated from each other.

The tree topology reflects a strong relationship among *L. bilineata* collected from different regions in South Africa and those collected in different localities in Italy, with one shared haplotype. Italian specimens of *L. bilineata* show lower levels of COI genetic differences than those of the South African specimens, suggesting a stronger bottleneck effect in the invasion history. At population level, the analyzed specimens of *L. daturaphila* span a large part of its native geographical distribution range (from Canada to Costa Rica), and should reflect its intraspecific variability. Interestingly, the South African and Australian subclade consists of haplotypes not present within the American subclade. However, the small number of available specimens does not permit any definitive conclusion about the origin of the invasion process to South Africa and Australia. 

As expected, both *L. bilineata* adults and larvae readily accepted *N. glauca* and *D. stramonium* as alternative hosts to *N. tabacum, P. peruviana* and *S. origanifolia*, but surprisingly, larvae did not feed on *N. glutinosa*, which was instead accepted by adults. These data are in agreement with those reported by other authors, in particular by Diaz et al. [12], who tested numerous Solanaceae and non-solanaceous species. Three non-solanaceae reported as hosts for *L. bilineata*, *H. annuus* (sunflower) [47,48,49], *B. vulgaris* (beet) [47], and *Ribes uvacrispa* (gooseberry) [50], were not accepted during our trials, either by adults or by larvae. We expected this result because the related reports are ambiguous. For example, the photograph of *L. bilineata* presented in [49] clearly shows a very different species of Chrysomelidae (possibly Galerucinae: Alticini), and the report of gooseberry in the host list in [50] could be due to possible confusion between the common names of *R. uvacrispa* (gooseberry) and *Physalis peruviana* (Cape gooseberry). Our data demonstrate for the first time that *R. uvacrispa* is not a host for *L*. *bilineata* and that some other solanaceous species (*N. glutinosa*, *B. arborea, C. annuum* and *P. hybrida*), not suitable for larval development, may be accepted by adults.

Overall, *L. bilineata* seems to be rather oligophagous, even if its alimentary preference does not strictly reflect the phylogeny of Solanaceae. Indeed, its hosts belong to two different subfamilies of Solanaceae, the Nicotianoideae and Solanoideae, but only three tribes within the latter (Datureae, Physaleae, Nicandreae) are attacked; moreover, differential preferences can be observed both at generic and tribal level. Indeed, while *N. tabacum* and *N. glauca* are suitable for larval development, *N. glutinosa* was not accepted by larvae; within Datureae, *Datura* spp. are among the main hosts of *L. bilineata*, while *B. arborea* is not accepted by larvae, and adults feed on it only in no choice tests. No species belonging to other tribes of Solanoideae (in particular Solaneae) were attacked, except glabrous varieties of *C. annuum* (tribe Capsiceae), on which adults caused minor lesions in no choice tests, probably just representing feeding probes, as well as on *P*. *hybrida* in the subfamily Petunioideae.

## 5. Conclusions

Hitherto, only 14 alien invasive leaf beetle species have been reported in Europe (mostly in southern countries), representing just a small percentage (0.9%) of the total European fauna of leaf beetles, but their number is continuously increasing. Indeed, to the list recently presented in [51], which includes the iconic case of *Leptinotarsa decemlineata* Say and the cryptogenic devastating species *Epitrix papa* Orlova-Bienkowskaja, as *E. similaris* (Gentner), three more alien invasive leaf beetles have been added in Europe during the last decade: *Ophraella communa* LeSage [52], *Monoxia obesula* Blake [53] and *Colasposoma dauricum* [54]. *Lema bilineata* is the first alien invasive Criocerinae reported in Europe. The invasion potential of *Lema* species studied in this paper is high, due to the mobility of adults, which may be also found on plants and goods different from their habitual ones. Examples are 20 interception episodes of *L. bilineata* adults from 1943 to 1975 on grapevine imported from Chile to the USA [3], and the interception of an adult in a pineapple crown at a port of entry in the UK [55]. Moreover, our data suggest that the potential of active dispersal of *L. bilineata* may be high. Indeed, if 2017 was the actual year of invasion in Italy, as results of earlier samplings suggest, the pest would be able to cover a distance of about 90 km in a single year. In addition, it is important to consider that the fecundity of *Lema* species is high, with a reported average oviposition of 1200 eggs during a period of 65 days for *L. bilineata* and 2700 for *L. daturaphila* [56]. The possibility of successful colonization of a new habitat is increased by the high diffusion of one of their preferred hosts, *S. origanifolia*, in Western Europe [57]. Present data provide diagnostic methods to enable efficient separation of *L. bilineata* from other similar species, in particular *L. daturaphila*, useful to prevent their accidental introduction into Europe. The latter species could have heavier impacts on agriculture, as it feeds on several cultivated Solanaceae in addition to tobacco. More extensive samplings, with specimens collected from different localities, should be undertaken to identify the source populations and the route of invasion of these two species.

## Figures and Tables

**Figure 1 insects-11-00295-f001:**
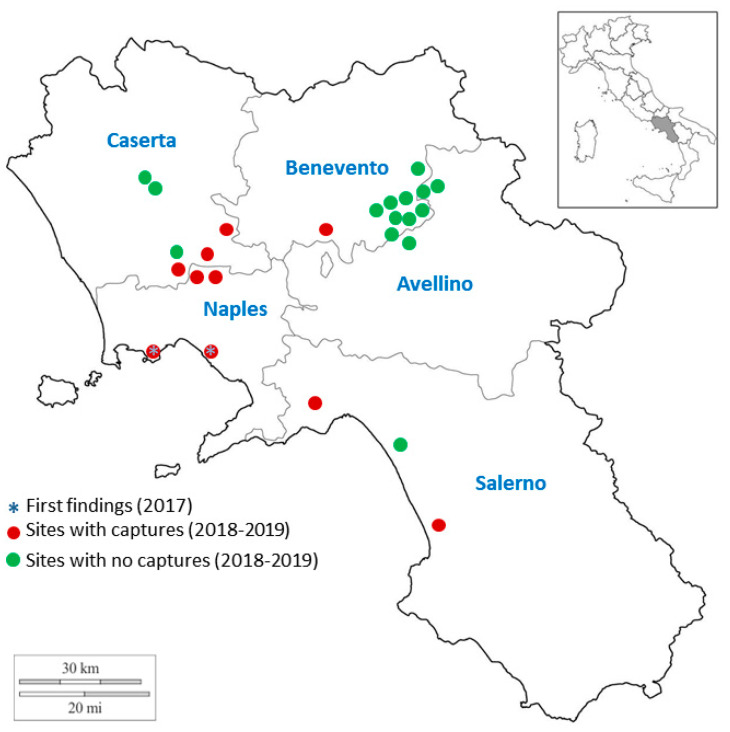
Distribution map for *Lema bilineata* in Campania (Italy) showing the first records in 2017 and subsequent samplings during 2018–2019.

**Figure 2 insects-11-00295-f002:**
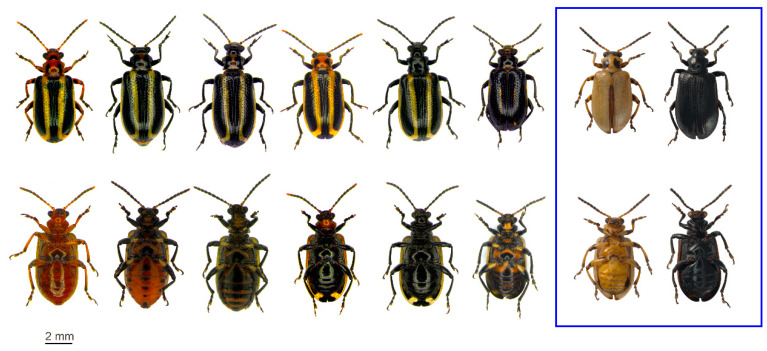
*Lema bilineata*: main color patterns in Italian specimens and extreme color forms in SANC specimens (within blue frame), in dorsal (upper row) and ventral (lower row) view.

**Figure 3 insects-11-00295-f003:**
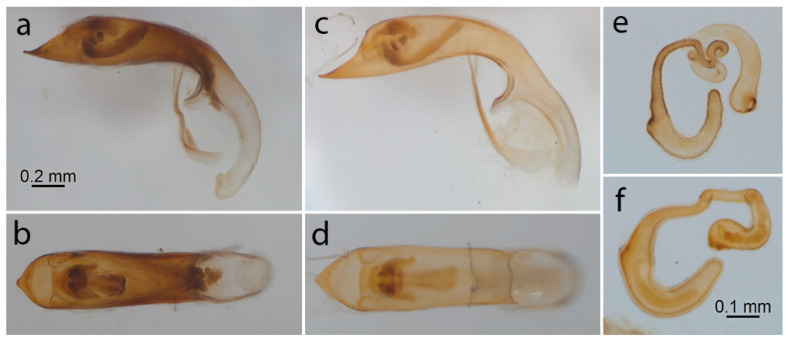
*Lema bilineata*: aedeagus in lateral (**a**) and dorsal (**b**) view; spermatheca (**e**). *Lema daturaphila*: aedeagus in lateral (**c**) and dorsal (**d**) view; spermatheca (**f**).

**Figure 4 insects-11-00295-f004:**
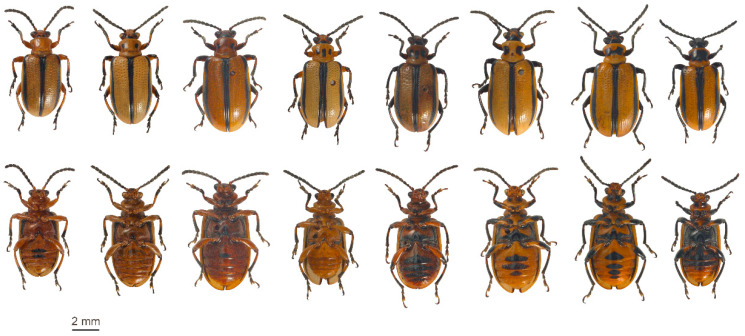
*Lema daturaphila*: main color patterns in South African specimens, in dorsal (upper row) and ventral (lower row) view.

**Figure 5 insects-11-00295-f005:**
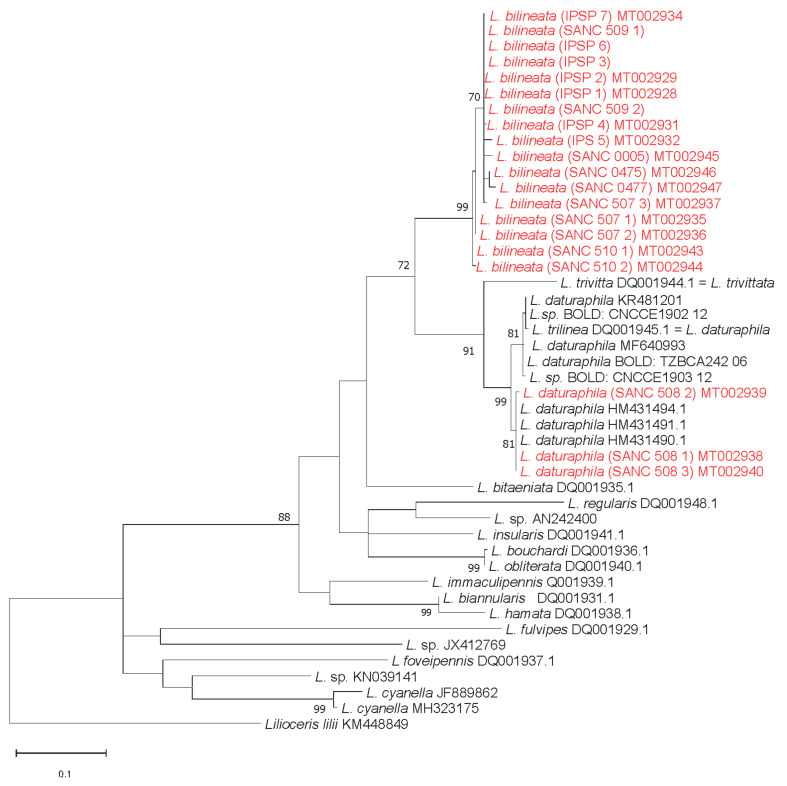
Phylogeny generated from maximum likelihood (ML) analysis of the cytochrome c oxidase I (COI) sequences from *Lema* species. The GenBank or Barcode of Life Data (BOLD) accession numbers of the sequences are reported. The nodes are marked by their ML bootstrap values, which are estimated from 10,000 bootstrap replicates. Only bootstrap values higher than 70% are shown. The novel sequences are in red.

**Table 1 insects-11-00295-t001:** List of novel sequenced specimens of *Lema bilineata* and *L. daturaphila*. SA: South Africa, I: Italy.

Voucher	*Lema* Species	Collection Site	Coordinates	Date	Storage	Host
SANC MCOL 0005	*L. bilineata*	SA: (Western Cape) Wilderness Ntl Park 10 km ESE George	34°00′ S 22°34′ E	20.xi.2000	frozen	not available
SANC MCOL 0475	*L. bilineata*	SA: (North West) Rustenburg	25°40′ S 27°15′ E	i.1996	dry	*Datura* sp.
SANC MCOL 0477	*L. bilineata*	SA: (Gauteng) Montana, Pretoria	25°40′ S 28°15′ E	18.i.1994	dry	*Physalis peruviana*
SANC MCOL 507-1	*L. bilineata*	SA: (Gauteng) Roodeplaat	25°36′ S 28°22′ E	09.xi.2018	EtOH	*Datura stramonium*
SANC MCOL 507-2	*L. bilineata*	SA: (Gauteng) Roodeplaat	25°36′ S 28°22′ E	09. xi.2018	EtOH	*Datura stramonium*
SANC MCOL 507-3	*L. bilineata*	SA: (Gauteng) Roodeplaat	25°36′ S 28°22′ E	09. xi.2018	EtOH	*Datura stramonium*
SANC MCOL 508-1	*L. daturaphila*	SA: (Gauteng) Roodeplaat	25°35′ S 28°20′ E	29.x.2018	EtOH	*Datura stramonium*
SANC MCOL 508-2	*L. daturaphila*	SA: (Gauteng) Roodeplaat	25°35′ S 28°20′ E	29.x.2018	EtOH	*Datura stramonium*
SANC MCOL 508-3	*L. daturaphila*	SA: (Gauteng) Roodeplaat	25°35′ S 28°20′ E	29.x.2018	EtOH	*Datura stramonium*
SANC MCOL 509-1	*L. bilineata*	SA: (Gauteng) Roodeplaat	25°36′ S 28°22′ E	05. xi.2018	EtOH	*Datura stramonium*
SANC MCOL 509-2	*L. bilineata*	SA: (Gauteng) Roodeplaat	25°36′ S 28°22′ E	05. xi.2018	EtOH	*Datura stramonium*
SANC MCOL 510-1	*L. bilineata*	SA: (Gauteng) Roodeplaat	25°36′ S 28°22′ E	05. xi.2018	EtOH	*Datura stramonium*
SANC MCOL 510-2	*L. bilineata*	SA: (Gauteng) Roodeplaat	25°36′ S 28°22′ E	05. xi.2018	EtOH	*Datura stramonium*
IPSP 1	*L. bilineata*	I: (Campania) Capaccio	40°27′ N 15°03′ E	27.vii.2018	fresh	*Nicotiana tabacum*
IPSP 2	*L. bilineata*	I: (Campania) Caserta	41°04′ N 14°20′ E	23.vi.2018	fresh	*Salpichroa origanifolia*
IPSP 3	*L. bilineata*	I: (Campania) Acerra	40°57′ N 14°20′ E	26.vi.2018	fresh	*Nicotiana tabacum*
IPSP 4	*L. bilineata*	I: (Campania) Cava de’ Tirreni	40°44′ N 14°42′ E	18.vii.2018	fresh	*Nicotiana tabacum*
IPSP 5	*L. bilineata*	I: (Campania) Montesarchio	41°02′ N 14°38′ E	01.viii.2018	fresh	*Nicotiana tabacum*
IPSP 6	*L. bilineata*	I: (Campania) Napoli	40°50′ N 14°14′ E	25.vii.2017	fresh	*Physalis peruviana*
IPSP 7	*L. bilineata*	I: (Campania) Portici	40°48′ N 14°21′ E	02.viii.2017	fresh	*Salpichroa origanifolia*

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
