# Peer review of "Morphological and Molecular Characterization of Lema bilineata (Germar), a New Alien Invasive Leaf Beetle for Europe, with Notes on the Related Species Lema daturaphila Kogan & Goeden"

_insects, 2020, doi:10.3390/insects11050295_

Round 1

Reviewer 1 Report

Dear authors,

your paper is of high importance to applied and general entomologists, not only in Europe but overall, since the topic of invasive species, their correct identification and their possible threat is crucial for effective pest control and for a comprehensive understanding of the biology of the respective species. I recommend unanimously publication of your paper.

I found only extremly few typos or other lapsus that I flagged on the attached pdf. I have two (possibly) major suggestions:

  • it would be very helpful for readers if you would compile a synoptic table of the characters that allow for telling apart the two (or three) species Lema bilineata, L. daturaphila, and possibly L. trivittata.
  • as far as I see, a positive proof that the studied species do not have a flagellum could only be reached by investigating the everted endophallus. The study by Matsumura et al. (2014) does not give a flagellum length for the Lilioceris spp., thus suggesting that they do not have a flagellum. However, in the cited papers by Berti & Rapilly (1976) and Düngelhoef & Schmitt (2006) the presence of a flagellum in Lilioceris spp. is shown. Therefore, either add a cautious phrase like "seemingly there is no flagellum" or something else to that effect, or present a photo of the everted endophallus showing the absence of a flagellum.

I do not insist in anonymity. If you wish, please feel free to contact me: michael.schmitt@uni-greifswald.de

Author Response

Please take note that the line out of brackets are relative to the corrected version, inside in the original one

Line 30 (Line 29). Added “, 1823”

Line46 (Line 45). Added “, 1970,”

Line 56 (Line 55). Corrected the sentence, added “. Furthermore”

Lines 78 and 86 (Lines 76 and 84). Corrected

Line 141 (Line 140). Removed “it”

Line 142 (Line 143). Fig. 1. A different symbol was selected, this way should be easier to spot and correctly identify the two sites near the coast, where the first samples were found.

Lines 194, 278 and 390 (Lines 192, 276 and 389). We slightly changed the sentences in a more cautious direction by adding “apparently there is no flagellum “.

Lines 391-392 (Line 389). We added “A definitive proof of the absence of flagellum could be gained through observation of everted endofallus [44]”.  We added this reference: Düngelhoef, S; Schmitt, M. Functional morphology of copulation in Chrysomelidae–Criocerinae and Bruchidae (Insecta: Coleoptera). Bonn Zool Beitr. 2005, 54.

Line 401 (Line 399). Added “they”

About the request  “it would be very helpful for readers if you would compile a synoptic table of the characters that allow for telling apart the two (or three) species Lema bilineata, L. daturaphila, and possibly L. trivittata” we think that the variability shown by the two Lema species could not be easily reported in a synoptic table, as the differences are mostly due to colour pattern. Therefore, we prefer to show the differences through the images.

Reviewer 2 Report

The paper by Monti and colleagues entitled “Morphological and molecular characterization of Lema bilineata (Germar), a new alien invasive leaf beetle for Europe, with notes on the related species Lema daturaphila Kogan & Goeden” presents a valuable work, with relevant taxonomical and ecological information on a recently recovered alien invasive species, Lema bilineata. The adopted methodologies are sound, and results reported in a clear and concise way; the manuscript is well written and deserve to be published in the present version with only minor changes.

Here below I have reported my minor suggestions:

Abstract

Line 14. I suggest the addition of “(Coleoptera: Chrysomelidae)” after leaf beetle

Line 17. I suggest the use of morphological features or characters instead of details

Line 21. Instead of “..photographic evidence.” I suggest to use “…macrographs.”

Line 22. Please use the full name of COI, at least the first time, Cytochrome c oxidase I

Introduction

Line 29. “the beetle called”  “the leaf beetle called”

Materials and Methods

Lines 58-59. I suggest the addition of “(Italy)”

Line 116. Please add citation for GTR  Lanave C, Preparata G, Saccone C, Serio G (1984) A new method for calculating evolutionary substitution rates. J Mol Evol 20: 86–93.

Results

Lines 142-144. I suggest to report the years in full, both in the figure legend and description.

Line 183. I suggest to include in figure 2 also one bigger high quality image of the species in dorsal view. The already shown morphs variability is fundamental, but also a bigger image of one specimen where morphological details can be appreciated is relevant too. The same observation can be done for figure 4.

Since the sampling allows it, I suggest to include the mean values of length and width (and their variability) of the specimens, for males and females.

After “:” capital is not required.

Line 311. I suggest the addition of “…K2P nucleotide …”

Line 314. Even here, I have the same observation of line 311. In addition I would like to suggest you to add throughout the text “nucleotide” before the word “distance”

Conclusions

Line 437. I think that a nice addition will be the inclusion of the names of all alien leaf beetles species reported in Europe, since in recent years their number seems increasing (Luperomorpha xanthodera Johnson and Booth 2004, Conti and Raspi 2007; Ophraella communa Boriani et al. 2013; Monoxia obesula Clark et al. 2014; Colasposoma dauricum Montagna et al. 2016)

References

Line 533. Change “-” with “–” as separator in the page number.

Author Response

Line 14 (Line 14). We added “(Coleoptera: Chrysomelidae)” after leaf beetle

Line 17 (Line 17). We changed “details” with characters

Line 22 (Line 21). We changed “..photographic evidence.” to  “…micrographic images.”, instead of macrograph. In our opinion macrograph is an image taken at a scale that is visible to the naked eye.

Lines 23 and 95 (Lines 22 and 93). We added the full name Cytochrome c oxidase I, both in the abstract and in Materials and methods

Line 30 (Line 29). Changed “the beetle called” with “the leaf beetle called”

Line 31 (Line 32). Added “Campania”

Line 61 (Lines 58-59). Instead of adding “(Italy)” here as suggested by the Referee2, we have specified “Italy, Campania” in the introduction, line 32

Line 117 (Line 116). Added citation for GTR+G+I as [34]. Added also in References: Lanave C, Preparata G, Saccone C, Serio G (1984) A new method for calculating evolutionary substitution rates. J Mol Evol 20: 86–93.

Lines 143 and 145 (Lines 142 and 144). Changed the years, reporting them in full, both in the figure legend and description, and added “(Italy)” after Campania

Lines 149 and 245 (Lines 148 and 243). About the comment “Since the sampling allows it, I suggest to include the mean values of length and width (and their variability) of the specimens, for males and females”: unluckily to take measures would require field trips and use of lab equipment, but both in this moment are restricted due to Covid 19 emergency. We added measures values of total length and width (and their variability) of museum specimens, which could not be sexed

Line 185 (Line 183). About the suggestion, “to include in figure 2 also one bigger high-quality image of the species in dorsal view. The already shown morphs variability is fundamental, but also a bigger image of one specimen where morphological details can be appreciated is relevant too. The same observation can be done for figure 4”, we preferred to add a bigger high-quality image of both species to be presented as supplementary material S1, cited in lines 148 and 244 (lines 147 and 242). In our opinion adding a bigger image in the fig 2 and 4 would result in an excessively small size of other samples, and furthermore presenting the same picture in two different size could be redundant.  

After “:” We changed capital with lowercase.

Lines 313 and 317 (Lines 311 and 314). We added  “…K2P nucleotide …”.

Line 439 (Line 437). We  included the most recent introduction of alien leaf beetles, reported in Europe, and their references (Luperomorpha xanthodera Johnson and Booth 2004, Conti and Raspi 2007; Ophraella communa Boriani et al. 2013; Monoxia obesula Clark et al. 2014; Colasposoma dauricum Montagna et al. 2016). We changed adding “Hitherto, only 14 invasive alien leaf beetle species have been reported in Europe (mostly in southern countries), representing just a small percentage (0.9%) of the total European fauna of leaf beetles, but their number is increasing continuously. Indeed, to the list recently presented in [51] which includes the iconic case of Leptinotarsa decemlineata Say and the cryptogenic devastating species Epitrix papa Orlova-Bienkowskaja, as E. similaris (Gentner)), three more invasive alien leaf beetles have been added for Europe during the last years: Ophraella communa LeSage [52], Monoxia obesula Blake [53] and Colasposoma dauricum [54]. Lema bilineata is the first invasive alien Criocerinae reported for Europe. The invasion potential …..”

Line 549 (Line 533). Changed “-” with “–” as separator in the page number